# Radiological Outcomes of Bone-Level and Tissue-Level Dental Implants: Systematic Review

**DOI:** 10.3390/ijerph17186920

**Published:** 2020-09-22

**Authors:** Saverio Cosola, Simone Marconcini, Michela Boccuzzi, Giovanni Battista Menchini Fabris, Ugo Covani, Miguel Peñarrocha-Diago, David Peñarrocha-Oltra

**Affiliations:** 1Oral Surgery Unit, Department of Stomatology, Faculty of Medicine and Dentistry, University of Valencia, 13, 46010 Valencia, Spain; miguel.penarrocha@uv.es (M.P.-D.); david.penarrocha@uv.es (D.P.-O.); 2Tuscan Stomatologic Institute, via Aurelia, 335, 55041 Lido di Camaiore, Italy; simosurg@gmail.com (S.M.); michela.boccuzzi@hotmail.it (M.B.); gbmenchinifabris@yahoo.it (G.B.M.F.); covani@covani.it (U.C.); 3Department of Stomatology, University of Studies Guglielmo Marconi, 44, 00193 Roma, Italy

**Keywords:** tissue-level, bone-level, dental implants, transmucosal, marginal bone loss, systematic review

## Abstract

***Background***: to assess the radiological marginal bone loss between bone-level or tissue-level dental implants through a systematic review of literature until September 2019. ***Methods***: MEDLINE, Embase and other database were searched by two independent authors including only English articles. ***Results***: The search provided 1028 records and, after removing the duplicates through titles and abstracts screening, 45 full-text articles were assessed for eligibility. For qualitative analysis 20 articles were included, 17 articles of them for quantitative analysis counting a total of 1161 patients (mean age 54.4 years) and 2933 implants, 1427 inserted at Tissue-level (TL) and 1506 inserted at Bone-level (BL). The survival rate and the success rate were more than 90%, except for 2 studies with a success rate of 88% and 86.2%. No studies reported any differences between groups in term of success and survival rates. Three studies showed that BL-implants had statistically less marginal bone loss (*p* < 0.05). Only one study reported statistically less marginal bone loss in TL-implants (*p* < 0.05). ***Conclusion***: In the most part of the studies, differences between implant types in marginal bone loss were not statistically significant after a variable period of follow-up ranged between 1 and 5 years.

## 1. Introduction 

Dental implants are the gold standard treatment to restore single edentulous space and partially or completely edentulous jaws because of their long-term success rate, the positive impact on patients’ quality of life, and the simplified modern surgical procedures with low morbidity [1,2].

Several factors can influence the preservation of hard and soft tissues around dental implants. Among these factors, the clinician’s experience, loading time, surgical protocol, implant neck configuration, implant-abutment connection, the insertion torque, and oral hygiene/maintenance protocols have been shown to influence the different outcomes of the implant therapy [3].

To reach the increasing patients’ needs for aesthetic results, low cost and fastest result, several factors must be taken into account before choosing the implant type and the protocol with the goal of a long survival and success rates of the implant-prosthetic rehabilitation. Among these factors, the clinician’s experience, the loading time, the type or surgery, the insertion torque, the oral hygiene maintenance protocols, the implant neck configuration and the implant-abutment connection, may influence the preservation of healthy peri-implant hard and soft tissues [4].

Clinical parameters (bleeding score and gingival index) and radiographic parameters (marginal bone loss) are used to evaluate the stability of the peri-implant soft and hard tissues.

The most used and objective clinical and radiological parameters to evaluate the stability of the peri-implant soft and hard tissue, so that the success of the rehabilitations, are respectively bleeding score, gingival index and marginal bone loss (∆MBL) [5]. Dental implants, after the healing period of 2–5 months, are anchored to the bone because of osseointegration. Traditionally, implants are two-pieces, so they are connected to the prosthetic rehabilitation through a transmucosal component, called abutment [6].

The early bone loss is observed after the connection of the abutment and when the prosthesis is loaded on the implant. It is well-known that there are a lots of factors to explain marginal bone resorption around dental implants such as: the occlusal trauma, biologic width establishment, gingival biotype, insertion torque of the implants, prosthesis loading timing, thickness of the remaining bone, type of surgery, primary stability, lack of bone to implant contact (BIC), bacterial colonization of the implant-abutment junction (IAJ), the macro and micro characteristic of abutment and the coronal portion of the fixture (shoulder/neck of the implant), and the position of the implant [7].

To avoid some of these disadvantages, Schroeder and co-workers introduced a “one-piece” implants to remove the contamination of the implant-abutment junction (IAJ) and to reduce the micromovements in the connection [8].

Nevertheless, one-piece implants have a difficult first surgery due to vertical dimension and due to the orientation of the remaining bone and the final prosthetic rehabilitation. One-piece implants must be inserted according to the final prosthesis position, not only considering hard and soft tissue availability. Moreover, if there are biomechanical complications, it may be more difficult it is not possible to remove to replace the abutment and final prosthesis. the abutment, instead all the implant must be removed. 

In modern literature, the term “one-piece implant” has modified its meaning. The new conception of “one-piece implants” regards both endosseous and transmucosal components, but the link with the abutment remains, located at increased distance from the bone, at tissue level [9].

To compare one-piece and two-pieces implants several clinical studies and some systematic review were performed in the last years.

Iglhaut and co-workers stated that the microgrooved surface could be associated with a longer connective tissue attachment and less bone resorption around implants [10].

Even though, considering the literature data, doubts still remain about the question: “What is the difference between one piece (bone-level) and two-piece (transmucosal) dental implants at single or multiple edentulous sites in terms of clinical and radiological outcomes during a long follow-up period?

Focusing on the literature until September 2019, the aim of the present systematic review was to identify whether there are relationships between different implants’ position (tissue level or bone level) and radiographic marginal bone loss in single or multiple rehabilitation, after at least 1-year of function. The terminology throughout the manuscript is various to be comprehensive for all synonymous which could be found in literature (e.g., tissue level/transmucosal/one piece versus bone level/two pieces implants).

## 2. Materials and Methods 

The present review has been conducted in accordance with the guidelines for Systematic Reviews and Meta-Analyses (PRISMA) [11].

Before starting the systematic review, a protocol has been developed and registered at PROSPERO, (International prospective register of systematic reviews, National Institute for Health Research, University of York, York, UK) with number: CRD42020157607.

This question follows the PICO (Population, intervention, comparison, outcomes) guidelines. The population (Population) was systemically healthy patients who (Intervention) received at least one implant and those implants that had been in place for at least one year. The Comparison in this type of studies was between two treatment groups according the level of implants: bone level and tissue level implants. The Outcome was the marginal bone loss.

The focused question was: are there any differences in terms of marginal bone loss in single or multiple rehabilitation between bone level implant and transmucosal/tissue level implant?

The rationale is based on the position of implant-abutment connection which could influence the healing process of the peri-implant tissues even after 1 or more years of follow-up, because of inflammation and bacterial infiltration in the micro-gap [12,13].

### 2.1. Search Strategy

The search was carried out independently by two authors and on four databases (MEDLINE, Embase, Inspec, and Cochrane Central Register of Controlled Trials) using synonyms as [(dental Implant OR abutment) AND (shoulder design OR implant abutment interface OR transmucosal OR bone-level OR scalloped implant OR sloped implant OR flat implant OR one-piece or two-pieces)].

The search was limited to articles in English. No restrictions on date of publication or follow-up period were applied when searching the first electronic databases to be as inclusive as possible. These databases were carried out until September 2019.

The exclusion criteria were applied after the electronic search. The bibliographies of all identified clinical included studies and relevant review articles were checked in order to identify other eligible articles related to the topic.

A complementary manual search that included a complete revision up to September 2019 was made of the following journals: Journal of Clinical Periodontology, Journal of Periodontal Research, Journal of Oral Science & Rehabilitation and Journal of Dental Research.

### 2.2. Study Selection and Eligibility Criteria

Randomized clinical trials (RCTs), case-control studies, comparative studies, and clinical trials comparing the clinical and/or radiological outcomes of different dental implant shoulder/neck position related to the crestal bone have been searched. The publications with the following inclusion criteria were selected:▪Comparison of different neck/shoulder position (One-piece vs. two-pieces or tissue-level or transmucosal vs. bone-level) of dental implants with at least 1-year follow-up after loading;▪Patients aged between 18 and 70 years old;▪Patients without severe systemic (e.g., recent cardiovascular event or tumoral pathology) or psychiatric disease;▪Clinical and radiological parameters measured were at least respectively bleeding on probing (BoP), and marginal bone loss (∆MBL);▪Only studies published in English.

The gingival recession was evaluated as a secondary outcome of interest in order to compare the possible association of one type of electric toothbrush with gingival recession prevalence. Reviews, letters, animal model, and vitro studies were excluded. Other exclusion criteria were:▪Studies included orthodontics patients;▪Studies included patients with disabilities;▪Studies included patients who are taking bisphosphonates;▪Studies comparing two or more different types of implant-abutment connections (e.g., switching platform) not focusing on position related to the bone;▪Studies comparing two or more different types of implant surgical technique with similar implant (e.g., one step surgery or two step) not focusing on position related to the bone;▪Studies comparing two or more different types of implant or abutment micro design;▪Studies comparing two or more different types of micro design of the implant neck or of the abutment;▪Final timepoint after less than 1 year after loading;▪Studies evaluating short-implants (in literature defined as implant <8.5 mm) [14];▪Studies analyzing implants and abutments used to retain removable prosthesis;▪Studies published before 1990;▪Duplicated studies or studies with different time points were included only one time with the longest duration.

### 2.3. Screening and Study Selection

Records identified through database searching were upload on End-Note (ISI Researchsoft 2001, Berkeley, CA, USA, http://www.endnote.com) to exclude the duplicates.

Then, titles and abstracts of all remaining articles were independently scanned by two reviewers following inclusion and exclusion criteria. Disagreements between authors were resolved after discussion by the intervention of a third author.

For studies appearing to meet the inclusion criteria, or for which there were insufficient information in the title and abstract to assess a clear decision, the full-text was obtained. The screening of full-text articles was performed by two reviewers independently to establish whether or not the studies met the inclusion and exclusion criteria. Disagreements were resolved by discussion of two authors. When resolution was not possible, a third reviewer was consulted.

Full-text rejected at this, or subsequent stages, were recorded in the table of excluded studies explaining reasons for exclusion.

All full-text articles meeting the inclusion criteria and assessed for eligibility were evaluated again by three authors to assess the quality of the methodology of each article and to perform data extraction. 

### 2.4. Quality Assessment (Risk of Bias of Included RCTs)

A quality assessment of the included studies was performed according to the Cochrane Handbook for Systematic Reviews of Interventions (version 5.1.0; updated March 2011 by Higgins and Green).

According to handbook guideline five main quality criteria were evaluated:Random sequence generation,Allocation concealment,Blinding of participants, personnel, and outcomes assessors,Incomplete outcome data,Selective outcome reporting.

Depending on the descriptions given for each main article of included studies, these criteria were rated as: low, unclear, or high risk of bias.

### 2.5. Quantitative Analysis

Mean marginal bone changes values were extracted from each study by one author (S.C.) and compared weighting parameters according to the number of implants for each study using a descriptive statistic. The number of implants was multiplicated by the number of MBL so that a study with a bigger number of implants value more than a study with a lower number of cases. The data were analysed using the T-test with a *p* < 0.05.

## 3. Results

The purpose of this review was to summarize the available evidences reported in literature of the included studies comparing the marginal bone loss of bone level (BL) versus tissue level implants (TL). Bone level changes between one-piece (TL) and two-piece (BL) dental implants. The combined search in four databases provided 1028 records (Figure 1).

After removing the duplicates using the software End-Note (ISI Researchsoft 2001) and the screening of title and abstract according to the relevance of the topic 45 articles remained. Following inclusion and exclusion criteria, the full-text of these article was obtained. In Table 1, excluded articles were reported with reasons, three of them (colored in grey) were excluded only from quantitative analysis, but not qualitative [15,16,17,18,19,20,21,22,23,24,25,26,27,28,29,30,31,32,33,34,35,36,37,38,39,40,41,42].

At the end of the study selection, a last revision was performed again by two authors and 17 articles were included for the final quantitative analysis (20 considering also qualitative analysis as reported in Table 2).

### 3.1. Qualitative Analysis

The data collected from each study were resumed in Table 3 [15,18,31,43,44,45,46,47,48,49,50,51,52,53,54,55,56,57,58,59].

In four studies the implants were positioned in the maxilla [18,43,53,58] in eight studies the implant rehabilitation involved the mandible [45,49,50,51,52,54,55,56] while in the other studies the patients received the implants in both jaws.

In two studis [43,58] the implants were inserted in the anterior region of the maxilla, in one study [53] the implants were positioned in the anterior region of the mandible, whereas in three studies [45,54,55] the implant treatment was performed in the posterior region of the mandible. In the majority of the studies, the patients were treated with the dental implants in both anterior and posterior region of maxillae.

All studies analyzed two types of implant systems: bone level implants and tissue level implants in different groups with different surgery and prosthetic protocols by different clinicians.

The parameter “Marginal Bone Level” was evaluated by the radiographic examination (intraoral radiography) in order to compare the changes in the bone level at the baseline and in the different time of follow-up. 

The timing of each follow-up varied considerably through the studies, from a first evaluation at a minimum of 3 months (Gulati 2013) to a maximum of 5 years [31,44,45,49,51,53,56,57,58], even if the overall follow-up ranged from 1 years to 3 years in the majority of the studies. 

The study included 1161 patients (mean age 54.4 years), who needed implants rehabilitation for mandibular and maxilla edentulism by fixed and removable prosthetic prostheses [53]. 

In total, 2933 implants were placed, 1427 according to the non-submerged protocols and 1506 according to the traditional submerged procedure. In both groups (submerged versus transmucosal group), the most used implants brands were the Branemark implants system Nobel Biocare AB, the ITI systems, the Astra Tech system, and the Straumann systems with exception for some studies as noticed in Table 3. 

For both implant systems, the fabrication of fixed prostheses has provided for single crowns and bridges in most cases. Some authors did not specify the prosthetic protocol and no information was given about the design of the framework except for [31,45,53,54,55,56].

In the present review, the only parameter used was marginal bone loss changes in the quantitative analysis because of the too wide variability of each studies in other clinical outcomes. The studies analyzed bleeding score did not reported any statically significant differences between groups.

The survival rate and success rate if reported were more than 90%, except for two studies [44,49] that had a success rate of 88% and 86.2% respectively. No studies reported any differences between groups in term of success and survival rates.

Three studies [45,48,57] showed that BL-implants had statistically less marginal bone loss compared with TL-implants (*p* < 0.05). Only one study [53] reported statistically greater peri-implant bone maintenance over time in TL-implants (*p* < 0.05).

In the most part of the studies, differences between implant types in marginal bone loss were not statistical neither clinically significant.

### 3.2. Quality Assessment

The methodological quality was assessed using the “Downs and Black Scale” and the “New Castle Ottawa Scale Cohort Studies” as suggested by the Cochrane Handbook [60].

The quality scores was graded as high for the studies with a score ≥ 24, medium for the studies with a score between 12 and 24 and low for the studies with a score ≤ 12 [61,62].

Two reviewers investigated the internal validity of the eligible studies and according to the quality assessment tool and the reported results in Figure 2, fourteen studies showed high quality [15,18,31,45,46,49,50,51,53,54,55,56,57,58] meanwhile the other studies showed a moderate quality [18,43,44,47,48,52,58].

Two authors investigated on the factors that could systematically affect the observations and the conclusions of the studies [63].

The two independent and calibrated authors assessed each single study, according as shown in Figure 3, papers were divided according to risk of bias in three categories: low risk, moderate risk, and high risk [63].

The tool items were scored as 1 if the item was considered fully fulfilled, as 0 if the item was clearly not fulfilled, and as 0.5 if the item was unclearly or only partially fulfilled.

Studies with a score ≤2.3 were considered high risk of bias, with a score between 2.4 and 4.6 as moderate risk, and with a score ≥4.7 were considered low risk of bias. 

The majority of the studies showed a low risk of bias, whereas six studies had a moderate risk of bias: [43,44,48,50,51,52].

### 3.3. Quantitative Analysis

Seventeen studies were selected for the quantitative analysis and the marginal bone loss comparison. The radiographic outcome refers to a total of 980 patients and 2260 implants: 1178 implants in BL-groups and 1082 implants in TL-groups.

Three studies reported marginal bone changes at 6 months, twelve studies at 1 year, five studies at 3 year, and seven studies at 5 year, only few studies reported the values at 3 months at 18 months, and finally at 2 year [46].

It had the following follow-up: 6 months, 12 months, 24 months, 36 months, and 60 months reporting better performances of BL-implants in the first time except for 60 months for which there are no radiological values for TL-implants.

The distribution of the mean bone level changes is presented in Table 3. The mean values of marginal bone loss (changes) were weighted considering the number of implants of each study, so that differences between the two intervention groups were calculated with a significance <0.05.

At 3 and 4 months the marginal bone loss was less in BL-groups, but the differences were not significative, plus only one study had such a short follow-up.

At 6 months the marginal bone loss was calculated in a total of 115 implants and it was lower in BL-groups than TL-groups, with a significance <0.05. This statistically significant difference was inverted at 12 months (less in TL than BL) with a sample of 1850 implants (*p* < 0.01). The follow-up at 12 months is the most representative of all included studies. The bone loss reported was again less in TL at 18 months, less in BL at 24 months and 36 months, but these three time-points were not statistically significant. 

After 60 months of follow-up the mean marginal bone loss was less in BL-group then in TL-group with a sample of 1069 implants (*p* < 0.01).

## 4. Discussion

This review gave clinicians an overall view of the topic to improve the knowledge of the marginal bone level changes after several years of follow-up, thus showing if different implant systems (bone level vs. tissue level) could affect bone resorption. 

Two recent systematic reviews and meta-analyses conducted by Sanz-Martín and colleagues analyzed all randomized controlled trials (RCTs) until 2016 that investigated macroscopic design, surface topography, and the manipulation of the abutment [64,65]. The authors reported no significant differences between these implants on peri-implant parameters. Only the abutment material had a significant impact on BoP values and ∆MBL. 

While Sanz-Martín and colleagues focused on the topic on abutment, other reviews studied the shoulder of the fixture. Starch-Jensen, Christensen, and Lorenzen [66] reported significantly more peri-implant marginal bone loss and higher BoP score in implants with a scalloped implant-abutment connection and not in the flat implant-abutment connection, despite their initial hypothesis, but their review included only three studies.

Also, Tallarico and colleagues, in another systematic review on the topic that also analyzed studies until 2016 (seven RCTs and five comparative studies), highlighted no significant evidence that the implant shoulder position/orientation and design offered improvements in clinical and radiological outcomes [67]. Nevertheless, they also included one-piece implants and they admitted that these results were limited because of the quality of available studies.

In the present analysis, three studies [18,43,59] reported slightly better values in marginal bone loss in BL-groups than TL but with no significance. Also no significant differences were found in the three studies [44,46,47] in which the marginal bone loss was slightly less in TL than in BL; while three studies reported statistically significant bone loss lower for BL than TL and only one study highlighted better radiological outcomes in TL. The other ten studies did not highlight any statistical differences.

Due to the heterogeneity of the included studies according to implant designs, sample sizes, patient parameters, time of evaluation and follow-ups, valid statistical comparisons are not possible, perhaps descriptive statistic for quantitative analysis was used.

The results of this review showed that most parts of the articles reported no differences between BL and TL according to bone loss, survival and success rate, or clinical outcomes. This observation is confirmed by the last review which reported a similar bone loss for both types of implants [68].

Despite the bone level changes being worse in TL-implants than BL-implants at 6 months of follow-up (*p* < 0.05) and 60-months of follow-up (*p* < 0.01), as reported in the results, the time-point the most representative of all quantitative analysis is 12-months because it has the larger sample of implants involving a major number of studies. In fact, 12 studies reported the radiological outcome at 12 months of follow-up, 7 studies at 60 months of follow-up, and only 3 studies at 6 months.

It could be reasonable to assume that the results at 12-months are more important for the number of implants and studies involved, nevertheless there are too many differences for each time-point. Moreover, [48] missed the radiological outcome of TL-implants at 60-months so only the comparison on the other time-point were analyzed. 

The results of the present review are limited because of the quality of data, the number of comparable available studies, and the wide variability, all of which could influence the final results, plus, the authors were not calibrated in the screening process.

Also, the different surgical protocols (one-stage or two-stages) may influence the bone changes, especially in the first period of healing and the prosthetic final rehabilitation is not standardized in the included articles, plus some studies did not report the specific information about the kind of prothesis.

Moreover, in the majority of the studies included in this review, the implants were inserted in mandible, but the bone quality and healing are different between upper and lower jaws.

It has been reported that the peri-implant tissues are more susceptible to inflammation than natural teeth [69]. Nevertheless, the definitions of survival rate/success/failure used in the literature do not necessarily reflect the patients’ chances of success or the function and aesthetics of the treatment because bleeding on probing (BoP), increasing of pocked probing depth (PPD), and other clinical outcomes are surrogates and the true link with the peri-implant tissues is questionable [70].

It has been reported that the implant type and the surgical protocol (bone level vs. tissue level) is correlated to the soft tissue bleeding response after probing (BoP) because of the presence of a chronic infiltrate at the implant-abutment interface of two-piece implants, attributed to the micro-gap between the implant and the abutment [71].

On the contrary, the peri-implant tissue around transmucosal implants has been reported to be inflammation-free, possibly because of the absence of a micro-gap reporting a lower prevalence of BoP [72,73]. According to the literature, in fact, there is a higher BoP prevalence detected in two-piece implants than in one-piece implants [74].

These studies, plus the biological rationale for the inflammation, have been pushing clinicians and researchers to assess if there are clinical differences in BoP and marginal bone changes in one- and two-piece dental implants. However, in more recent articles, these clinical and radiological differences between types of implants are not statistically significant and are reported probably because of the more efficient platform switching, the surface of the neck, and the type of abutment material [75,76]. 

Moreover, several authors have reported a lack of correlation between clinical outcomes (PPD and BoP) and crestal bone loss around implants [77,78].

A retrospective study on 4591 implants in 2060 patients with 10-year follow-up reported that, while BoP was very commonly detected on implants (40% of implants in the cohort study), only 3% of these implants had more than 1 mm of marginal bone loss. The study concluded that the minimal bleeding on probing in implants was not correlated with marginal bone loss and therefore probing healthy implants was not recommended [79].

Even if the BoP level is reported to be more frequent in BL implants compared to TL one as a marker of local inflammation in two-pieces implants, the latest systematic review conducted by Paul and Petsch [80] reported no differences between these types of implants, in fact the clinical examination of BoP around dental implants is not completely validated as a clinical outcome to evaluate the bone loss.

In the present review, a statistically less marginal bone loss was reported at one-year follow-up in TL-implants, but after five years of follow-up the BL reported statistically better results. The heterogeneity of the results in the different studies and the oscillation between BL and TL bone loss according to the follow-up are possibly due to other confounding factors such as implant micro surfaces, implant shape, and the implant-abutment connection to prevent bacterial infiltration [81,82].

Implant dentistry needs more prospective studies with more standardized characteristic considering, the bone quality, the site position, the type of loading in order to analyze the bone loss differences associated to one or two-piece implants.

Some patients, especially with chronic disease, may benefit from transmucosal implants because of the lack of bacterial leakage in the implant-abutment connection, but no evidence of long-term effect on bone loss is reported.

Nowadays, the focus should be shifted to the morphology and geometry of the implant neck. This could improve connective and bone stability and guide bone healing, especially in the period immediately after surgery [83].

## 5. Conclusions

In the present review, no evidence was found for differences in marginal bone loss or implant survival rate between bone-level and transmucosal dental implants after a period of follow-up variable from 12 to 60 months. It could be concluded that many other clinical and surgical variables influence marginal bone level and implant survival. More homogenous clinical trials with larger samples are needed to support these conclusions and to give more precise clinical indications.

## Figures and Tables

**Figure 1 ijerph-17-06920-f001:**
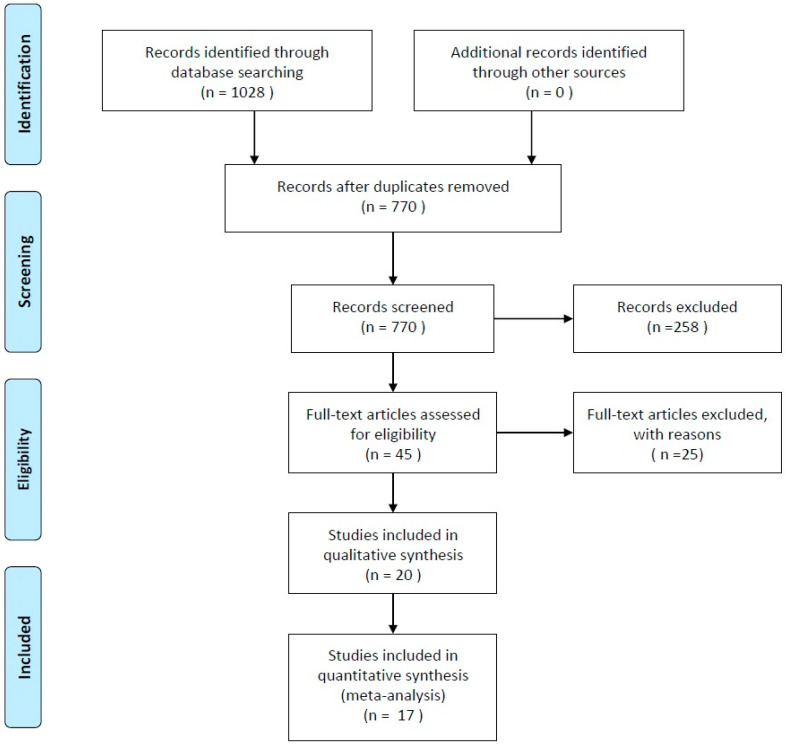
Flow chart diagram (2009) of search strategy adapted from PRISMA.

**Figure 2 ijerph-17-06920-f002:**
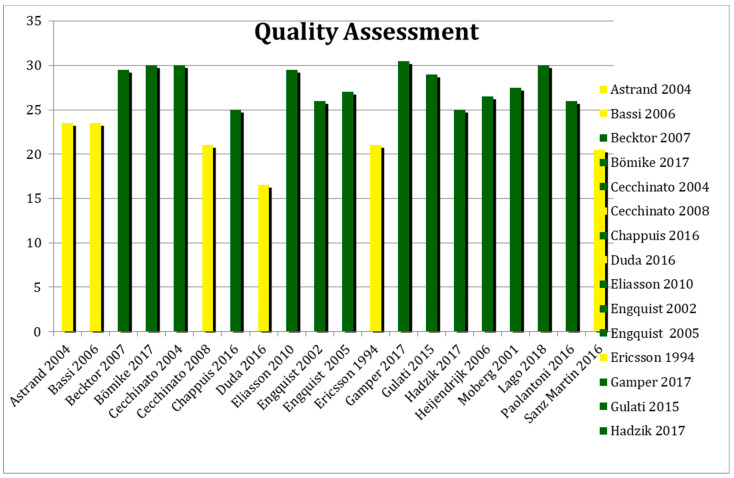
Quality assessment of the studies.

**Figure 3 ijerph-17-06920-f003:**
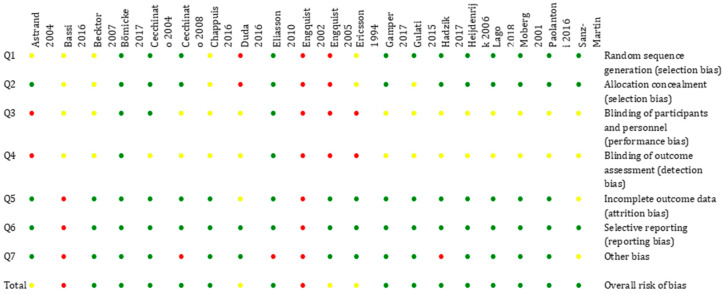
Risk bias word: Risk bias word: red–high risk; yellow–unclear; risk green-low risk.

**Table 1 ijerph-17-06920-t001:** Excluded articles after full-text screening; the articles marked in grey rows were included for qualitative analysis, but not quantitative one [15,16,17,18,19,20,21,22,23,24,25,26,27,28,29,30,31,32,33,34,35,36,37,38,39,40,41,42].

No.	References	Exclusion Motivation
1	Becktor JP, Isaksson S, Billström C. A prospective multicenter study using two different surgical approaches in the mandible with turned Brånemark implants: conventional loading using fixed prostheses [15]	Excluded for the quantitative analysis: The parameter “marginal bone level” was not clearly reported.
2	Bratu EA, Tandlich M, Shapira L. A rough surface implant neck with microthreads reduces the amount of marginal bone loss: a prospective clinical study [16]	Studies comparing 2 or more different types of implant or abutment micro design.
3	de Siqueira RAC, Fontão FNGK, Sartori IAM, Santos PGF, Bernardes SR, Tiossi R. Effect of different implant placement depths on crestal bone levels and soft tissue behavior: a randomized clinical trial [17]	Studies comparing 2 or more different types of implant or abutment micro design.
4	Chappuis V, Bornstein MM, Buser D, Belser U. Influence of implant neck design on facial bone crest dimensions in the esthetic zone analyzed by cone beam CT: a comparative study with a 5-to-9-year follow-up [18]	Excluded for the quantitative analysis: excluded because it reports median, not mean value of MBL.
5	Chien HH, Schroering RL, Prasad HS, Tatakis DN. Effects of a new implant abutment design on peri-implant soft tissues [19]	Studies comparing 2 or more different types of micro design of the implant neck or of the abutment.
6	Cosyn J, Sabzevar MM, De Wilde P, De Rouck T. Two-piece implants with turned versus microtextured collars [20]	Studies comparing 2 or more different types of implant or abutment micro design.
7	Ebler S, Ioannidis A, Jung RE, Hämmerle CH, Thoma DS. Prospective randomized controlled clinical study comparing two types of two-piece dental implants supporting fixed reconstructions—results at 1 year of loading [21]	Studies comparing 2 or more different types of implant surgical technique with similar implant (e.g., one step surgery or two step) not focusing on position related to the bone.
8	Esposito M, Trullenque-Eriksson A, Blasone R, et al. Clinical evaluation of a novel dental implant system as single implants under immediate loading conditions—4-month post-loading results from a multicentre randomised controlled trial [22]	Studies comparing 2 or more different types of implant surgical technique with similar implant (e.g., one step surgery or two step) not focusing on position related to the bone.
9	Hof M, Pommer B, Strbac GD, Vasak C, Agis H, Zechner W. Impact of insertion torque and implant neck design on peri-implant bone level: a randomized split-mouth trial [23]	Studies comparing 2 or more different types of implant or abutment micro design.
10	Herrero-Climent M, Romero Ruiz MM, Díaz-Castro CM, Bullón P, Ríos-Santos JV. Influence of two different machined-collar heights on crestal bone loss [24]	Studies comparing 2 or more different types of implant surgical technique with similar implant (e.g., one step surgery or two step) not focusing on position related to the bone.
11	Judgar R, Giro G, Zenobio E, et al. Biological width around one- and two-piece implants retrieved from human jaws [25]	Studies comparing 2 or more different types of implant surgical technique with similar implant (e.g., one step surgery or two step) not focusing on position related to the bone.
12	Khorsand A, Rasouli-Ghahroudi AA, Naddafpour N, Shayesteh YS, Khojasteh A. Effect of Microthread Design on Marginal Bone Level Around Dental Implants Placed in Fresh Extraction Sockets [26]	Studies comparing 2 or more different types of implant surgical technique with similar implant (e.g., one step surgery or two step) not focusing on position related to the bone.
13	Khraisat A, Zembic A, Jung RE, Hammerle CH. Marginal bone levels and soft tissue conditions around single-tooth implants with a scalloped neck design: results of a prospective 3-year study [27]	Studies comparing 2 or more different types of implant-abutment connections (e.g., Switching platform) not focusing on position related to the bone.
14	Kim JJ, Lee DW, Kim CK, Park KH, Moon IS. Effect of conical configuration of fixture on the maintenance of marginal bone level: preliminary results at 1 year of function [28]	Studies comparing 2 or more different types of implant surgical technique with similar implant (e.g., one step surgery or two step) not focusing on position related to the bone.
15	Kütan E, Bolukbasi N, Yildirim-Ondur E, Ozdemir T. Clinical and Radiographic Evaluation of Marginal Bone Changes around Platform-Switching Implants Placed in Crestal or Subcrestal Positions: A Randomized Controlled Clinical Trial [29]	Studies comparing 2 or more different types of implant surgical technique with similar implant (e.g., one step surgery or two step) not focusing on position related to the bone.
16	Marconcini S, Giammarinaro E, Toti P, Alfonsi F, Covani U, Barone A. Longitudinal analysis on the effect of insertion torque on delayed single implants: A 3-year randomized clinical study [30]	Studies comparing 2 or more different types of micro design of the implant neck or of the abutment.
17	Moberg LE, Köndell PA, Sagulin GB, Bolin A, Heimdahl A, Gynther GW. Brånemark System and ITI Dental Implant System for treatment of mandibular edentulism. A comparative randomized study: 3-year follow-up [31]	Excluded for the quantitative analysis:The parameter “marginal bone level” was not clearly reported.
18	Nóvoa L, Batalla P, Caneiro L, Pico A, Liñares A, Blanco J. Influence of Abutment Height on Maintenance of Peri-implant Crestal Bone at Bone-Level Implants: A 3-Year Follow-up Study [32]	Studies comparing 2 or more different types of micro design of the implant neck or of the abutment.
19	Ormianer Z, Duda M, Block J, Matalon S. One- and Two-Piece Implants Placed in the Same Patients: Clinical Outcomes After 5 Years of Function [33]	It is the topic of the present review but it is a case series.
20	Pellicer-Chover H, Peñarrocha-Diago M, Peñarrocha-Oltra D, Gomar-Vercher S, Agustín-Panadero R, Peñarrocha-Diago M. Impact of crestal and subcrestal implant placement in peri-implant bone: A prospective comparative study [34]	Studies comparing 2 or more different types of implant surgical technique with similar implant (e.g., one step surgery or two step) not focusing on position related to the bone.
21	Peñarrocha-Diago MA, Flichy-Fernández AJ, Alonso-González R, Peñarrocha-Oltra D, Balaguer-Martínez J, Peñarrocha-Diago M. Influence of implant neck design and implant-abutment connection type on peri-implant health. Radiological study [35]	Studies comparing 2 or more different types of implant or abutment micro design.
22	Pozzi A, Agliardi E, Tallarico M, Barlattani A. Clinical and radiological outcomes of two implants with different prosthetic interfaces and neck configurations: randomized, controlled, split-mouth clinical trial [36]	Studies comparing 2 or more different types of implant surgical technique with similar implant (e.g., one step surgery or two step) not focusing on position related to the bone.
23	Pozzi A, Tallarico M, Moy PK. Three-year post-loading results of a randomised, controlled, split-mouth trial comparing implants with different prosthetic interfaces and design in partially posterior edentulous mandibles [37]	Studies comparing 2 or more different types of implant-abutment connections (e.g., Switching platform) not focusing on position related to the bone.
24	Sanz-Martin I, Vignoletti F, Nuñez J, et al. Hard and soft tissue integration of immediate and delayed implants with a modified coronal macrodesign: Histological, micro-CT and volumetric soft tissue changes from a pre-clinical in vivo study [38]	It is a study on animal model (Dog).
25	Shin YK, Han CH, Heo SJ, Kim S, Chun HJ. Radiographic evaluation of marginal bone level around implants with different neck designs after 1 year [39]	Studies comparing 2 or more different types of implant or abutment micro design
26	Tan WC, Lang NP, Schmidlin K, Zwahlen M, Pjetursson BE. The effect of different implant neck configurations on soft and hard tissue healing: a randomized-controlled clinical trial [40]	Studies comparing 2 or more different types of implant surgical technique with similar implant (e.g., one step surgery or two step) not focusing on position related to the bone.
27	Weinländer M, Lekovic V, Spadijer-Gostovic S, Milicic B, Wegscheider WA, Piehslinger E. Soft tissue development around abutments with a circular macro-groove in healed sites of partially edentulous posterior maxillae and mandibles: a clinical pilot study [41]	Studies comparing 2 or more different types of micro design of the implant neck or of the abutment.
28	Wittneben JG, Gavric J, Belser UC, et al. Esthetic and Clinical Performance of Implant-Supported All-Ceramic Crowns Made with Prefabricated or CAD/CAM Zirconia Abutments: A Randomized, Multicenter Clinical Trial [42]	Studies comparing 2 or more different types of micro design of the implant neck or of the abutment.

**Table 2 ijerph-17-06920-t002:** All studies included for the qualitative analysis. The 3 studies in grey rowed were excluded from the quantitative analysis as explained in the text [43,44,45,46,47,48,49,50,51,52,53,54,55,56,57,58,59].

	Studies Qualitative Analysis	Study Design	Patients Sample	Number of Implants (BL/TL)	Mean Age Range of the Sample	Type of 6 Implants BL; TL	Type of Prosthetic Restoration	Success Rate BL/TL	Survival Rate BL/TL	Follow-Up
1	Astrand P. [43]	Prospective Randomized Comparative Multicenter Study	28	73/77	61.7 ± SD range: 36–76	BL: Branemark TL:ITI	Fixed Partial Bridges	/	100%	12 Months; 36 Months;
2	Bassi M. [44]	Prospective Clinical Study	133	66/67	60 ± 11 range: 29–75	BL: I-Fiz EVO Conical; TL: Shiner EVO Conical;	52 Single Crown/3 Overdenture/70 Bridges	88%	100%	60 Months;
3	Becktor. [15]	Prospective Multicenter Study	80	206/198	TL: 63.5 ± 9.1 Range: 47–89 BL: 65.5 ± 9.4 Range: 44–84	Branemark System Nobel Biocare AB	Fixed Prosthetic Dentures		97.6%/91.4%	6 Months; 12 Months; 36 Months;
4	Bömicke W. [45]	Randomized Controlled Trial Study	38	19/19	TL: 54.37 ± 14.62 BL: 51.51 ± 13.96	Nobel Biocare AB	Single Zirconia Crown	/	100%/94.7%	12 Months; 36 Months;
5	Cecchinato D. [46]	Multicenter Randomized Controlled Crinical Trial	84	171/153	51.6	Astra Tech	Fixed Prosthetic dentures	/	>98%	12 Months; 24 Months;
6	Cecchinato D. [46]	Multicenter Randomized Controlled Crinical Trial	84	171/153	51.6	Astra Tech	Fixed Prosthetic Dentures	/	>98%	24 Months; 60 Months;
7	Chappuis V. [18]	Comparative Study	61	20/41	TL: 38.8 Range: 24–72 BL: 41.7 Range: 24–60	Straumann	Single Crown	/	/	60 Months;
8	Duda M. [48]	Non Randomized Retrospective Study	33	29/24	TL: 42.5 BL: 53.6	Q Implants Trinon Titanium GmbH		/	100%/91.7%	6, 12, 36 Months; 60 Months;
9	Eliasson A. [49]	prospective clinical study	29	84/84	65	DBA Paragon	Full arch ISFP	86.2%	99.4%	12 Months; 60 Months;
10	Engquist B. [50]	Controlled Prospective Study	82	113/80	TL: 65 BL: 64	Branemark System Noble BIocare AB	Fixed Prosthetic bridges	/	97.5%/93.2%	12 Months;
11	Engquist B. [51]	Controlled Prospective Study	108	110/106	64.9	Branemark System Nobel Biocare AB	Fixed Prosthetic Bridges with Cantilever	/	100%/100%	12 Months; 36 Months;
12	Ericsson I. [52]	Longitudinal Study	11	33/30	61 Range: 42–72	Branemark System	Fixed Prosthetic Bridges	/	/	12 Months; 18 Months;
13	Gamper F.A. [53]	Randomized Controlled Clinical Trial Study	60	86/65	TL: 47.5 ± 15 BL: 55.8 ± 14	BL: Branemark system Nobel Biocare AB TL: Straumann	Removable Prosthetic Prostheses/Screw Retained prostheses/cemented prostheses	/	98.9%/96.6%	60 Months;
14	Gulati M. [54]	Prospective Randomized Comparative Study	19	10/10	TL: 28.22 ± 3.27 BL: 27.20 ± 2.78 Range: 23–33	Adin Dental Implant System	Screw-Retained Porcelain Fused to Metal Prosthesis	/	/	3 and 6 Months;
15	Hadzik J. [55]	Clinical Study	13	16/16	TL: 46.3 BL: 45.9 Range: 20–63	BL: Osseospeed TX, Astra tech TL: RN SLActive^®^, Straumann	Cemented Crowns	/	100%	6 Months;
16	Heijdenrijk K. [56]	Prospective Randomized Study	60	38/38	58 ± 11	Unknown	Overdenture with Clip Attachment	/	/	12, 24, 36, 48, and 60 Months;
17	Lago L. [57]	Randomized Clinical Trial	100	102/100	50.5 Range: 25–70	Straumann	Single Crowns		96.1%/98%	12 and 60 Months;
18	Moberg [31]	Randomized Prospective Study	40	103/106	BL: 62.6 ± 7.0 Range: 44.2–75.2 TL: 64.0 ± 6.8 Range: 40.2–77.2	BL: Branemark System Nobel Biocare AB TL: ITI system	Screw Prosthetic Bridges	97.9%/96.8%	/	6 Months; 12 Months; 36 Months;
19	Paolantoni G. [58]	Randomized Controlled Clinical Trial Study	65	29/45	53 ± 4	Thommen Medical AG	Single Crowns		100%	60 Months;
20	Sanz-Martin I. [59]	Prospective Randomized Controlled Clinical Study	33	18/15	Unknown	BL: Branemark System Nobel Biocare AB TL: Strumann	Group 2 Piece: SCs-4FDPs Group 1 Piece: SCs–4FDPs	/	/	12 Months;
	Total		1161	2933						3–60 Months;

**Table 3 ijerph-17-06920-t003:** The mean values of marginal bone loss (changes) in relation to the number of implants of each study with corresponding *p*-values.

Mean Value Marginal Bone Changes #	Bone Level Implant	Tissue Level	Significance (*p* < 0.05)
3 Months	0.19	0.28	/(Only Gulati 2013)
6 Months	0.33	0.42	0.0169 * (3 studies)
*n* = 115	*n* = 65	*n* = 50	
12 Months	0.25	0.18	0.0000 * (12 studies)
*n* = 1850	*n* = 971	*n* = 879	
18 Months	0.05	0.04	/(Only Ericsson 1994)
24 Months	0.18	0.24	0.1907 (2 studies)
36 Months	0.45	0.48	0.5031 (5 studies)
48 Months	1.4	1.6	/(Only Heijdenrijk 2006)
60 Months	0.29	0.38	0.0050 * (7 studies)
*n* = 1069	*n* = 576	*n* = 493	

# The mean values of marginal bone loss (changes) are weighted considering the number of implants of each study. * The T-test reported significance with *p* < 0.05.

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
