# Peer review of "Radiological Outcomes of Bone-Level and Tissue-Level Dental Implants: Systematic Review"

_ijerph, 2020, doi:10.3390/ijerph17186920_

Round 1

Reviewer 1 Report

This paper was a systematic review about the radiological differences between Bone Level implants and Tissue Level Implants.

The authors well clarified the study limitations and well concluded that a lot of clinical and surgical variables can influence marginal bone level changes and implant survival, so that more homogenenous clinical trials with larger samples are needed to give more appropriate clinical indications.

 some typographical errors:

line 143 were instead than ere;

line 164-169: five criteria.....:the first two criteria were reported without numebring.

Author Response

Dear doctor, thank you for the precious help to review the manuscript.
I followed you advices and we corrected the typographical errors:

line 143 were instead than ere;

line 164-169: five criteria.....:the numbers.

Best regards,

Dr. Saverio Cosola

Reviewer 2 Report

Dear Authors,

Thank you for submitting the manuscript entitled "Radiological outcomes of bone-level and tissue-level 2 dental implants: systematic review."

I read it with a great interest and would like to congratulate you and your team for writing such a relevant systematic review.

The paper is well written and I just have a couple of comments/suggestions that I hope will enhance the clarity of this manuscript. Therefore, I recommend publication of this work after the Authors address the minor points listed below.

Abstract

Abstract should be follow the instructions for Authors requested by IJERPH, such as:

  • 200 words maximum;
  • the style of structured abstracts, but without headings: 1) Background; 2) Methods; 3) Results and 4) Conclusion.

Introduction

In my opinion, Introduction should be shortly elaborated, especially when Authors assert that several factors must be taken into account before choosing the implant type. Please, when list these factors at lines 43 – 46, provide an appropriate reference for each factor.

Moreover, since implant-supported rehabilitation of the edentulous ridge require adequate volume and integrity of the alveolar bone, this factor should be considered in your list and references such as “Barone, A.; Covani, U. Maxillary alveolar ridge reconstruction with nonvascularized autogenous block bone: clinical results. J Oral Maxillofac Surg 2007 ,65, 2039-46” and “Grassi, F.R.; Grassi, R.; Vivarelli, L.; Dallari, D.; Govoni, M.; Nardi, G.M.; Kalemaj, Z.; Ballini, A. Design Techniques to Optimize the Scaffold Performance: Freeze-dried Bone Custom-made Allografts for Maxillary Alveolar Horizontal Ridge Augmentation. Materials 2020, 13, 1393” should be enclosed.

Lines 47- 49:  When Author assert “The most used and objective clinical and radiological parameters to evaluate the stability of the peri-implant soft and hard tissue, so that the success of the rehabilitations, are respectively bleeding score, gingival index and marginal bone loss (ΔMBL)”, an appropriate reference should be added.

Search strategy

Line 99: Please replace “independent” with “independently”.

Study selection and eligibility criteria

Why is the age cut-off at 70 year olds? I would provide more information to justify this threshold for exclusion.

Table 3

Please, in the table legend specify what grey row means, as reported in table 2.

Please, throughout the manuscript replace "prothesis" with "prosthesis".

Please, divide Conclusions from Abbreviation (lines 418 – 419).

Please, add the Author contribution paragraph.

Author Response

Dear reviewer thank you for your time and your precious suggestions that I hope bring this manuscript to be published. I agree with your advices and I modified the text accoring to it.

Best regards,

Dr. Saverio Cosola

Reviewer 3 Report

Title and Introduction

  1. Title must correlate with research question and findings. While the title claims radiological outcomes, the only outcome assessed is the marginal bone loss (MBL) post-implant placement.
  2. In the introduction, the authors mention about one piece implants and correlate them with currently available implants. This is a bit confusing and distracts from the setting for the review. Moreover, the authors use alternating terminology throughout the manuscript, which should be avoided. Eg. – Tissue level/transmucosal – Tissue level/Bone level - One piece/ two piece..???
  3. In the introduction, authors claim three measurable outcomes of implant rehabilitation, namely bleeding score, gingival index and MBL. Why then compare only the MBL in the review? Why not include clinical parameters to make the review comprehensive?
  4. While both the aim and the focused questionof the review are correlating, “whether there are relationships between different implants’ position (tissue level or bone level) and radiographic marginal bone loss in single or multiple rehabilitation, after at least 1-year of function”, the results of the review do not reflect the same.

Methods

  1. Why choose to manual search in only 4 specific periodontology related journals? What about the other journal related to implant dentistry and restorative dentistry, which have reported studies related to post-implant outcomes?
  2. In inclusion/exclusion criteria, do you mean mini-implants or short implants? Clearly specify the measurement as length or diameter of the implant. Because mini-implants and short-implants are totally different clinical entities.
  3. What was the reason behind exclusion of screw retained implants in the review?
  4. Authors have to clarify, whether included studies had standardized radiographic measurements? If so, how? What was the criteria for measuring MBL radiographically?
  5. Mention abbreviated names of authors in parentheses, to identify the reviewers for study selection and the third author who helped in resolution of disagreements.
  6. Please elaborate quantitative analysis of weighted mean calculation and how T-test was applied to the weighted mean.

Results

  1. Table 2 – mention that grey shaded studies are excluded only for quantitative synthesis as an inclusive foot note to the table.
  2. Table 3 - includes data on screw retained prostheses and overdentures. While exclusion criteria states the opposite.
  3. Why not include demographic details such as maxilla/mandible or anterior/posterior jaws in table 3, instead of describing them in paragraph text?
  4. Please include citations from the bibliography, for the studies mentioned in tables 2 and 3.
  5. Overall results need reformatting – avoid writing in multiple small paragraphs.
  6. Qualitative analysis describes only the demographic data. There is no information regarding how MBL may be affected by tissue level or bone level implant placement, or any other confounding factors. Also, there is no mention about the correlation between clinical and radiological outcomes, qualitatively. This is the most significant take-away report that would interest potential readers and clinicians.
  7. Data from the figures and tables need not be replicated in text. Only the references need to cited from bibliography. Eg. Studies with low risk of bias… Studies with high quality of reporting…etc.
  8. Table 4 to be renamed as Figure/Graph 1. The same figure/graph has to include legend for the 2 different colors used. Moreover, the figure/graph appears distorted in the PDF review version.
  9. Table 5 to be renamed as Figure/Graph 2. The same figure/graph has to include legend for the different colors used. Moreover, the figure/graph appears distorted in the PDF review version.
  10. Table 6 to be renamed as Table 4. Moreover, the comparative data presented in the table is a bit confusing. What is the role of the “N” mentioned in between rows of different follow up periods? Does it indicate the numbers of implants quantified per follow up period or is it based on the studies included? Authors should try to redefine the table in an easily understandable format for all readers.
  11. Data from table 6 has again been replicated in the text.

Discussion

  1. The authors should direct their discussion on how MBL could be affected by bone level or tissue level implant placement? What is the significance of MBL on long term implant performance, considering the fact that implant survival was very high irrespective of MBL, in all the reviewed studies?

General Comments

  1. The effect of several confounding variables on the outcomes remains unanswered throughout the review. For example, does the site of placement really affect MBL? Because bone level implants are more preferred in the anterior esthetic zone when compared to tissue level implants that are a choice in the posterior jaws. Similarly, is there an anatomic difference between mandible and maxillary implants on the MBL? Also, does a single stage or 2 stage surgical protocol affect MBL, irrespective of bone or tissue level implants. What is the role of cement or screw retained implants on the MBL? Is there a difference in MBL between single implant restorations, overdenture implants and implant supported prosthetic bridges? What is the role of age and gender on the MBL? For all the above confounders, if there is no quantitative data available to report, at least a qualitative review should have been discussed.
  2. The entire manuscript need to be revised for errors pertaining to English language, grammar and punctuation. When describing numbers with decimals use period (.) as a separator, instead of comma (,). Majority of the sections of the manuscript are written as one or maximum two points per paragraph, and the whole manuscript has several paragraphs. This style is not suited for scientific publication. Authors should endeavor to revise the same.

Author Response

Dear reviewer, thank you for your precious advices, I modified the manuscript according your comments. In the introduction, we claimed three measurable outcomes of implant rehabilitation, as bleeding score, gingival index and MBL but then we compared only MBL because of the available data in each study.

We choose manual search in only 4 specific periodontology related journals because for the others the online search was comprehensive.

Best regards,

Reviewer 4 Report

In relation to the manuscript entitled "Radiological outcomes of bone-level and tissue-level  dental implants: systematic review" which I received for review.

The authors completed a systematic review of the literature without time restrictions to evaluate the radiographic marginal bone loss suffered by two types of dental implant approaches (bone-level and tissue-level). 

The topic is relevant because can affect clinical practice decisions, these two approaches are currently used in the dental field and much discussion has been generated around the outcomes of both approaches. Therefore, a review that synthesizes the current knowledge is required. 

Although the review was exhaustive, and there is scientific merit in the findings, there are some major drawbacks that require corrections and clarifications before publication.

-General comments to the manuscript

The manuscript English style and grammar must corrected. There are present typos and misspelled words along the manuscript, legend to tables and tables content.

The references should be checked and adapted all to the journal style.

-Comments to the abstract

Page 1. Line 17. Please edit the following sentence " To assess any differences on marginal bone loss" and replace by " To assess the radiographic marginal bone loss"

Page 1. Lines 23 and 24. Please edit the following sentence "A total of 1161 patients (mean age 54,4 years) and 2933 implants were observed, 1427 23 (Tissue-level) and 1506 (Bone-level)." and replace by "A total of 1161 patients (mean age 54,4 years) and 2933 implants, including 1427 inserted at tissue-level (TL) and 1506 inserted at bone-level (BL). "

Page 1. Lines 29 to 33. Please replace and rewrite  the following conclusions "Conclusion: Despite to the peri-implant 29 tissue around transmucosal implants has been reported to be inflammation-free because of the  absence of bacterial infiltration in the micro-gap between the fixture and abutment, no clinical and  radiological differences were highlighted between groups from the included studies after a variable 32 period of follow-up ranged between 1 to 5 years."

The conclusions must be specific to answer the aims of the study, differences in marginal bone loss between bone level and tissue level implants. Everything else can be included in the discussion section. In addition, the conclusions in your abstract, differ from the conclusions in the body of the manuscript. 

-Comments to the introduction

Given that this is not a dental journal. The authors must provide in the introduction section a detailed explanation of the marginal bone levels, what is expected from the bone levels around an implant after one year, what the bone level represents,  and why its stability is important for the long-term success and survival of the implant.

Page 1 and page 2. Lines 40 to 49. Please edit the following two paragraphs to reduce wording and for better clarity.

"To reach the increasing patients’ needs for aesthetic results, low cost and fastest result, several factors must be taken into account before choosing the implant type and the protocol with the goal of a long survival and success rates of the implant-prosthetic rehabilitation.  

"Among these factors, the clinician’s experience, the loading time, the type or surgery, the insertion torque, the oral hygiene maintenance protocols, the implant neck configuration and the implant-abutment connection, may influence the preservation of healthy peri-implant hard and soft  tissues. The most used and objective clinical and radiological parameters to evaluate the stability of the  peri-implant soft and hard tissue, so that the success of the rehabilitations, are respectively bleeding score, gingival index and marginal bone loss (ΔMBL)."

and replace by "Several factors can influence the preservation of hard and soft tissues around dental implants. Among these factors, the clinician's experience, loading time, surgical protocol, implant neck configuration, implant-abutment connection, the insertion torque, and oral hygiene/maintenance protocols have been shown to influence the different outcomes of the implant therapy." Please add the missing references

"Clinical parameters (bleeding score and gingival index) and radiographic parameters (marginal bone loss) are used to evaluate the stability of the peri-implant soft and hard tissues."  Also please add the missing references. 

Page 2. Lines 63 to 67. Please add the missing references to the following paragraph "Nevertheless, one-piece implants have a difficult first surgery due to vertical dimension and due  to the orientation of the remaining bone and the final prosthetic rehabilitation. One-piece implants 64 must be inserted according the final prothesis position, not only considering hard and soft tissue availability. Moreover, if there are biomechanical complications, it is not possible to remove the abutment, instead all the implant must be removed.

The authors navigate the introduction trough a series of paragraphs without logical connection, and supporting references are missing for many paragraphs and sentences. The authors did not describe the bone level and tissue level protocols in the introduction. The authors should explain what a tissue level implant, and a bone level implant are.   There are specific implant designs to be placed at bone or tissue level, but the clinician can also decide to what level the implants are going to be inserted. The authors must include in the introduction updated and enough references about the marginal bone level changes suffered by tissue-level and bone-level implants. An English speaker expert should review grammar, typos, and writing style. Supporting references must be added and updated.

-Comments to materials and methods

Page 3. Lines 106 to 108. Why the exclusion criteria were applied after the electronic search? Could this have affected the validity of your results by influencing the articles selection and biasing your search? Please clarify and discuss in the discussion section 

Page 3. Lines 113 to 115. Given that the authors included in the search RCTs, case-control studies, cohort studies (comparative studies), and clinical trials, With this broad search, the significance of the findings is reduced, the comparisons are not possible, and the extrapolation is not possible unless the results are grouped by type of study.

Besides unclear distinction between case series and cohort studies could result in inconsistent study selection and unjustified exclusions. Moreover, some studies have not been classified by their original authors and could be mislabeled in the systematic review. (Please see Ref. Mathes and Pieper. BMC Med Res Methodol. 2017;17:107) 

Please clarify in the discussion section

Page 3. Lines 122 and 123. A description of how the marginal bone loss was evaluated in each of the included studies should be incorporated in the materials and methods and results section. The landmarks differ in many studies, therefore, comparison is not possible. Please provide the results in descriptive statistics tables. Also, a definition for Delta Marginal Bone Loss) should be provided for improving  the readers understanding.

Page 3. Lines 125 and 126. In relation to the following sentence "The gingival recession was evaluated as a secondary outcome of interest in order to compare the possible association of one type of electric toothbrush with gingival recession prevalence.: Why the authors included electric toothbrush? or gingival recession? These variables are not part of the aims of this study. And they are too far away from the radiological evaluation which is the main goal of the study? Please clarify or correct.

Page 3. Line 139. Please correct "mini-implant" and replace by "short-implant"

Page 3. Line 143. Please edit the following sentence "Studies with results published more than once ere included only one time" and replace by "Duplicated studies or studies with different time points" If only one study is included should be that with the longest duration. 

Page 4. Line 144. In the screening and study selection, Did the authors that participated in the screening and selection process were calibrated? if so, what type if calibration? if calibration was not completed, please explain why?

Page 4. Lines 165 to 169. Please number all the criteria from 1 to 5. (numbers 1 and 2 of the five criteria are missing)

Page 4. Lines 173 and 174 in the quantitative analysis section. In the discussion section, please include a paragraph that explain your approach. Statistical comparisons are not possible between the included studies. Variability in implant designs, sample sizes, patient parameters, time of evaluation and follow-ups, preclude comparisons. However, the authors could input the descriptive statistics of the included studies. 

-Comments to results

Page 4. Lines 177 and 178. First paragraph of the results section. The authors used different aims in the abstract, at the end of the introduction section, and here. Please clarify, and unify your aims in all the sections. Furthermore the authors include here the one piece versus two piece dental implants concept. This creates confusion for the readers. This reviewer recommends to change this sentence by "The aim of the review was to compare the marginal bone loss of bone level versus tissue level implants."

Page 5. Lines 210 to 214. Please check the writing style and grammar of this paragraph

Pages 5, 6, and 7 in table 2.The reasons for exclusion here in table 2, must be the same that the authors described in the exclusion criteria section. Nothing else should be added, OR the exclusion criteria section should be modified. To allow the exclusion reasons presented in table 2.

Page 8. Lines 220 to 221. In relation to the following sentence "Possible disagreement was solved through the involvement of the third review with expertise in Implantology and oral surgery." Please delete. Is redundant and was described in the materials and methods section.

Page 9 and 10. Table 3. In the type of implants column. Please include the type of implants of each brand that the authors compared.  In the Astra tech, Straumann, Adin, Thommen, that information is missing. 

Please provide an additional table with all the reported marginal bone loss per each one of the included studies. The landmarks used by the original authors of each study used for the evaluation of the marginal bone loss in the radiographic analysis are missing and must be included. The implant surface characteristics for all the implants compared in the included studies are missing and must be included. This factors must be included in a new table. Then table 6 will makes sense.

-Comments to discussion

None

-Comments to conclusions

Please remove the following sentence from the conclusions paragraph and move it to the discussion section.

"Some patients, especially with chronic disease, may benefit from transmucosal implants because of the lack of bacterial leakage in the implant-abutment connection, but no evidence of long-term effect on bone loss is reported." Because this was not the aim of your study and can't be extrapolated from your results just can be fitted in the discussion section. 

-Comments to references

Please check the references, add the missing references to the references list, and adjust and edit as per the journal style. 

Author Response

(The authors gave the same response as above.)

Round 2

Reviewer 4 Report

The authors corrected the manuscript following all the reviewers recommendations. Therefore, the new version is significantly improved compared with the original.

The introduction now provides adequate background for the readers understanding, the materials and methods are better organized and can be reproduced, the results are presented appropriately in the text, and the graphics, figures, and tables supplement the information accurately. The discussion uses sound literature; the conclusions are clear and supported by the findings.

There is still room for improvement in the writing style. The authors should correct the English style,  and improve grammar and spelling errors.